# Efficient Transmit Antenna Subset Selection for Multiuser Space–Time Line Code Systems

**DOI:** 10.3390/s21082690

**Published:** 2021-04-11

**Authors:** Sangchoon Kim

**Affiliations:** Department of Electronics Engineering, Dong-A University, 37, Nakdong-Daero 550beon-gil, Saha-Gu, Busan 604-714, Korea; sckim@dau.ac.kr; Tel.: +82-51-200-7705

**Keywords:** antenna selection, multiple input multiple output (MIMO), precoding, zero forcing (ZF), minimum mean square error (MMSE), multiuser, space–time line code

## Abstract

We consider the problem of the efficient transmit antenna subset (TAS) selection for maximizing the signal-to-interference-plus-noise ratio (SINR) of multiuser space–time line code (MU–STLC) systems. The exhaustive search for optimal TAS selection is impractical since the total number of transmit antennas increases. We propose two efficient TAS selection schemes based on the Woodbury formula. The first is to incrementally select $N_{S}$ active transmit antennas among the available $N_{T}$ transmit antennas. To reduce the complexity of the incremental selection scheme, the Woodbury formula is employed in the optimization process. The second is to perform the decremental strategy in which the Woodbury formula is also applied to develop the low-complexity TAS selection procedure for the MU–STLC systems. Simulation results show that the proposed incremental and decremental TAS selection algorithms offer better alternatives than the existing greedy TAS selection algorithm for the MU–STLC systems. Furthermore, in terms of bit error rate, the proposed minimum mean square error decremental TAS selection algorithm turns out to outperform the existing greedy algorithm with significantly lower computational complexity. Finally, we analyze the detection SINR penalty experienced from TAS selection and the analytical quantity is shown to be well matched with simulation results.

## 1. Introduction

Multiple-input multiple-output (MIMO) transmission techniques have been employed as an integral part of present-day communication systems to improve the overall radio link capacity and reliability [1,2,3,4]. Space–time block code (STBC) has been used as one of the general MIMO transmission strategies when no channel state information (CSI) is available at the transmitter [5,6,7,8]. It spreads over time and over space (transmit antennas). STBC is an effective way to exploit the potential of MIMO systems because it is a very simple code requiring a low encoding and decoding complexity. Although it does not require multiple antennas at the receiver, the use of multiple receive antennas offers extra diversity gain and array gain.

Recently, a space–time line code (STLC) in [9] was presented as a new transmission method having full rate and diversity. In the STLC scheme, two information symbols are encoded by channel gains coming from multiple receive antennas and are sent consecutively in time. The STLC transmission is a dual version of Alamouti STBC [6], based on the symmetric CSI and antenna configurations. The STLC assumes the knowledge of the full CSI at the transmitter (CSIT), whereas the STBC requires the CSI at the receiver (CSIR). The STLC scheme is also advantageous because of its low complexity encoding and decoding procedure. Owing to its implementation simplicity, the STLC transmission technique has been used for various communication systems. For example, they include massive MIMO and multiuser systems [10,11], two-way relay systems [12], and machine learning-based blind decoding [13].

In the basic STLC scheme, multiple transmit antennas are not required at the transmitter. In case multiple transmit antennas are present, STLC benefits from additional diversity gain and array gain. In [14], a new multiuser (MU)–STLC scheme is designed to support simultaneous transmission of multiple STLC signals for multiple users through preprocessing at the transmitter. Moreover, a transmit antenna subset (TAS) selection problem for the proposed MU–STLC system has been initially investigated for performance enhancement. Firstly, an exhaustive search-based optimal TAS selection algorithm that maximizes the detection signal-to-interference-plus-noise ratio (SINR) is presented with tremendously huge complexity. To alleviate the complexity problem, an SINR-based greedy TAS selection algorithm is also proposed at the cost of performance degradation. However, it requires an $N_{R}\times N_{R}$ matrix inverse operation in each greedy step, where $N_{R}$ denotes the total number of receive antennas and is given as the product of the number of multiple users and the number of receive antennas per user. When the total number of receive antennas is large, its computational complexity is huge. Additionally, TAS selection in [14] does not guarantee optimal performance. Furthermore, the effects of TAS selection on the SINR performance have not been sufficiently studied. Recently, the performance of the STLC systems with TAS selection has been analyzed and evaluated in [15]. However, it considers only the single-user model. To the best of our knowledge, no other study, except for [14], has been previously made for effective TAS selection in the MU–STLC systems.

Antenna subset selection at the transmitter and/or receiver has been researched extensively in many kinds of MIMO systems [14,15,16,17,18,19,20,21,22,23,24,25]. It has been employed to improve the performance and reliability over those achievable with wireless communication systems without antenna subset selection. It can be also performed to reduce the number of expensive radio frequency (RF) chains while preserving spatial diversity gains. It is shown in [23,24] that to reduce the number of RF chains in linearly precoded multiuser MIMO systems and zero-forcing (ZF)-based precoded spatial modulation (PSM) MIMO systems, respectively, decreasing the number of active transmit antennas by TAS selection always degrades the bit error rate (BER) performance. Recently, two efficient TAS selection algorithms for PSM-based massive MIMO systems have been developed in [25]. However, the conventional various TAS selection algorithms presented in [25] and other studies on different MIMO systems are unsuitable for the MU–STLC systems owing to different transmission schemes. The reason for this unsuitability is mentioned in Section 3.1 after the system model of MU–STLC with TAS selection is described. It should be noted that one key issue in the antenna subset selection for the STLC transmission scheme is the optimal design of a proper selection criterion.

In this paper, two efficient TAS selection schemes that have a better tradeoff between transmission performance and computational complexity are proposed for MU–STLC by exploiting incremental and decremental strategies, respectively, combined with the Woodbury formula. First, the Woodbury formula is utilized to reduce significantly the complexity of the conventional greedy TAS selection algorithm proposed in [14]. Second, we propose a decremental TAS selection scheme in which the Woodbury formula is also used for enormous complexity reduction. For decremental selection, the MU–STLC transmitter uses preprocessing matrices based on ZF and minimum mean square error (MMSE) senses. It is shown that the proposed decremental TAS selection algorithm based on MMSE can offer near-optimal bit error rate (BER) performance with low computational complexity. Furthermore, we show that reducing the number of activated transmit antennas through TAS selection always degrades the SINR and BER performance. The detection SINR loss experienced from TAS selection is analytically obtained. Simulation results demonstrate that the detection SINR penalty agrees with the analytical one. Without BER simulations for a large number of transmit antennas, such as massive MIMO, we can anticipate how much BER performance is degraded in terms of signal-to-noise ratio (SNR) owing to reducing the number of activated transmit antennas through TAS selection.

The main contributions of this study are summarized as follows:The effective TAS selection algorithms based on the incremental and decremental methods combined with the Woodbury formula have been designed for the MU–STLC systems;The computational complexity of the proposed TAS selection algorithms is analyzed and compared to the optimal one and the previous TAS selection scheme of [14]. The complexity comparison proves the efficiency of the proposed algorithms;The asymptotic received SINR loss is analytically provided and verified by simulation results.

The rest of the paper is organized as follows. The system model of MU–STLC with TAS selection is described in Section 2. In Section 3, we propose incremental and decremental TAS selection algorithms to offer low complexity. In Section 4, the computational complexity of the proposed algorithms is analyzed and compared with that of the conventional greedy TAS selection scheme. In Section 5, a simulation-based comparison of the BER performance of the proposed algorithms and the previous method is provided. Concluding remarks are drawn in Section 6.

Notations, we use lower-case and upper-case boldface letters for vectors and matrices, respectively. Superscripts ${}^{*}$, ${}^{T}$, and ${}^{H}$ denote the complex conjugate, transposition, and Hermitian transposition, respectively. The notations Tr(⋅) and ${\left(\cdot \right)}^{-1}$ denote the trace and the inverse of a matrix, respectively. E[⋅], |⋅|, and ${\left\|\cdot \right\|}_{F}$ stand for the expectation, the absolute value, and the Frobenius norm, respectively. $I_{n}$ and $0_{n}$ denote the n×n identity matrix and the n×n matrix with all zero elements, respectively. X(:,k) indicates the *k*-th column vector of a matrix X. X(:,[1:(k−1)  (k+1):end]) stands for the submatrix remained by deleting the *k*-th column vector in a matrix X. $blkdiag\left\{\;Q_{2},\cdots ,Q_{2}\right\}$ returns a block diagonal matrix whose diagonal matrices are $Q_{2},\cdots ,Q_{2}$.$CN(0,\sigma_{}^{2})$ means a complex normal distribution with a zero mean and variance $\sigma_{}^{2}$.

## 2. System Model of MU–STLC with TAS Selection

We consider a downlink $N_{T}-\mathrm{by}-2K$ MU–STLC system with $N_{T}$ transmit antennas and K users. Each user has two receive antennas for STLC [9,14,15]. Thus, the total number of receive antennas is 2K. The transmitter is equipped with only $N_{S}(K\leq N_{S}\leq N_{T})$ RF transmission units. Thus, it is assumed that $N_{S}$ transmit antennas are selected from $N_{T}$ antennas. Let $x_{k,t}$ be the *t*-th transmitted symbol of the *k*-th user, with $E[x_{k,t}x_{k,t}^{*}]=\sigma_{x}^{2}$. Then, the MU–STLC signal matrix is defined as (1)$$U=[u_{1}\;\;u_{2}]=W_{S}X\in C^{N_{S}\times 2}$$
where $u_{t}={\left[u_{1,t}\;u_{2,t}\;\cdots \;u_{N_{S},t}\right]}^{T}\in C^{N_{S}\times 1}$, t=1,2, and X=${\left[X_{1}\;X_{2}\;\cdots \;X_{K}\right]}^{T}\in C^{2K\times 2}$ with
(2)$$X_{k}=\left[\begin{array}{cc} x_{k,1} & x_{k,2}\\ -x_{k,2}^{*} & x_{k,1}^{*} \end{array}\right]\in C^{2\times 2},\;k=1,2,\cdots ,K$$
and $W_{S}\in C^{N_{S}\times 2K}$ is the MU–STLC precoding matrix for all users such that ${\left\|W_{S}\right\|}_{F}^{2}=1$.

The received signals with TAS selection are then represented as
(3)$$\left[\;r_{1}\;\;r_{2}\right]=H_{S}U+Z\in C^{2K\times 2}$$
where $r_{t}={\left[\;r_{1,t}\;r_{2,t}\;\cdots \;r_{K,t}\right]}^{T}\in C^{2K\times 1}$. Here, $r_{k,t}={\left[\;\;r_{k,t}^{1}\;r_{k,t}^{2}\right]}^{T}$$\in C^{2\times 1}$ is the received signal vector where $r_{k,t}^{n}$ is the received signal at the *n*-th receive antenna of the *k*-th user at time *t*. $H_{S}\in C^{2K\times N_{S}}$ denotes the channel submatrix obtained by selecting $N_{S}$ columns from the full channel matrix $H=[\;h_{1}\;h_{2}\;\cdots \;h_{N_{T}}]\in C^{2K\times N_{T}}$. Here, $h_{m}$, $m=1,2,\cdots ,N_{T}$, is a channel vector between the *m*-th transmit antenna and all users, which is static for t=1 and t=2, and whose elements are independent and identically distributed (i.i.d.) circularly symmetric complex Gaussian random variables with zero mean and unit variance. $Z={\left[Z_{1}\;Z_{2}\;\cdots \;Z_{K}\right]}^{T}\in C^{2K\times 2}$ with $Z_{k}=[\;z_{1}\;z_{2}]\in C^{2\times 2}$ and $z_{t}={\left[\;z_{1,t}\;z_{2,t}\right]}^{T}\in C^{2\times 1}$, t=1,2, is an independent and identically distributed (i.i.d.) additive white Gaussian noise (AWGN) matrix whose elements are the zero-mean circular complex white Gaussian noise component of variance of $\sigma_{z}^{2}$.

For the MU–STLC decoding, the received signal matrix of (3) is re-expressed in a linear form as [14]
(4)$$r≜{\left[r_{1}^{T}\;r_{2}^{H}\right]}^{T}=\left[\begin{array}{c} H_{S}W_{S}\\ {\left(H_{S}W_{S}\right)}^{*}Q_{2K} \end{array}\right]x+z\in C^{4K\times 1}$$
where
(5)$$x≜{\left[x_{1,1}\;x_{1,2}\;\cdots \;x_{k,1}\;x_{k,2}\;\cdots \;x_{K,1}\;x_{K,2}\right]}^{T}$$
(6)$$Q_{2K}=blkdiag\left\{\;Q_{2},\cdots ,Q_{2}\right\}\in R^{2K\times 2K}$$
(7)$$Q_{2}=\left[\;\begin{array}{cc} 0 & -1\\ 1 & 0 \end{array}\right]$$ and $z\in C^{4K\times 1}$ is the AWGN vector with $E[\;z\;z^{H}]=\sigma_{z}^{2}\;I_{4K}$.

By the simple STLC combining procedure at the receiver described in [14], the multiuser combined-STLC received signal vector can be given as
(8)$$y={\tilde{H}}_{S}{\tilde{W}}_{S}\;x+z^{\prime }\in C^{2K\times 1}$$where
(9)$${\tilde{H}}_{S}≜[H_{S}\;\;Q_{2K}^{T}H_{S}^{*}]\in C^{2K\times 2N_{S}}$$
(10)$${\tilde{W}}_{S}≜{\left[W_{S}^{T}\;\;Q_{2K}^{T}W_{S}^{H}\right]}^{T}\in C^{2N_{S}\times 2K}$$
and the combined AWGN vector $z^{\prime }\in C^{2K\times 1}$ follows the distribution $CN(\;0_{2K},\;2\sigma_{z}^{2}\;I_{2K})$. Here, the MU–STLC precoding matrix ${\tilde{W}}_{S}$ can be given by
(11)$${\tilde{W}}_{S}={\tilde{\beta }}_{S}{\tilde{H}}_{S}^{H}\;{\tilde{V}}_{S}$$
where the power normalization factor related to the selected TAS is given as
(12)$${\tilde{\beta }}_{S}=\frac{1}{{\left\|\;{\tilde{H}}_{S}^{H}\;{\tilde{V}}_{S}\right\|}_{F}}$$
and ${\tilde{V}}_{S}$ is determined by ZF and MMSE precoders, respectively, as
(13)$${\tilde{V}}_{ZF,S}={\left(\;\;{\tilde{H}}_{S}^{}{\tilde{H}}_{S}^{H}\right)}^{-1}$$
(14)$${\tilde{V}}_{MMSE,S}=\left({\tilde{H}}_{S}^{H}{\tilde{H}}_{S}^{H}+\frac{2K\sigma_{z}^{2}}{\sigma_{x}^{2}}I_{2K}\right)-1$$


## 3. TAS Selection Algorithms

In this section, we first present previous optimal and suboptimal greedy TAS selection algorithms for the MMSE-precoded MU–STLC systems. Then, the popular Woodbury formula is exploited to obtain a suboptimal incremental SNR-based TAS selection method with more reduced complexity. Furthermore, we propose a decremental TAS selection algorithm based on ZF and MMSE criteria. Finally, an efficient algorithm for decremental TAS selection is developed by using the Woodbury formula.

### 3.1. Optimal Exhaustive Search-Based and SINR-Greedy-Based TAS Selection Algorithms

It is easily shown that maximizing the received SINR of the MMSE-precoded MU–STLC systems is equivalent to maximizing the term ${\tilde{\beta }}_{S}^{2}$. Hence, the TAS selection scheme for the MU–STLC systems can be expressed as
(15)$$S_{opt}=\underset{S\in \;\left\{\;S_{n},n=1,2,\cdots ,C\;\left(N_{T},N_{S}\right)\;\right\}}{\mathrm{arg max}}{\tilde{\beta }}_{S}^{2}$$
where $S_{n}$ is the *n*-th enumeration of the set of all available TASs. Here, $C(N_{T},N_{S})$ is the total number of combinations of selecting $N_{S}$ transmit antennas out of $N_{T}$ antennas. Then, the optimal TAS selection algorithm for the MMSE-precoded MU–STLC system is described as [14].
(16)$$S_{opt}=\underset{S\in \;\left\{\;S_{n},n=1,2,\cdots ,C\;\left(N_{T},N_{S}\right)\;\right\}}{\arg\;\min}Tr\left[\;{\tilde{V}}_{MMSE,S}\right]$$

It should be pointed out that in the ZF precoding case, the previous works of [24,25] use the minimization of $Tr\left[{\left(H_{S}^{}H_{S}^{H}\right)}^{-1}\right]$ for TAS selection in the PSM systems, whereas this work for the MU–STLC systems is based on the optimization of $Tr\left[{\left({\tilde{H}}_{S}^{}{\tilde{H}}_{S}^{H}\right)}^{-1}\right]$, where ${\tilde{H}}_{S}^{}$ is defined by (9). Thus, the TAS selection algorithms presented in [24,25] are unsuitable for TAS selection in the MU–STLC systems. Due to the difference between two channel matrices of $H_{S}^{}\in C^{2K\times N_{S}}$ and ${\tilde{H}}_{S}^{}\in C^{2K\times 2N_{S}}$, the Woodbury formula should be applied differently to the development process of the low-complexity algorithm, and thus, the succeeding efficient TAS selection algorithms proposed for the MU–STLC systems in this work are distinct from those in [24,25]. Furthermore, note that most of the studies on antenna selection, including [16,18,23], are based on the channel $H_{S}^{}$, not ${\tilde{H}}_{S}^{}$. That is why they are inappropriate for direct use in the MU–STLC systems.

Obviously, the exhaustive search algorithm to solve (16) requires $C(N_{T},N_{S})$ matrix inverse operations, whose computational complexity is tremendous, especially when the number of all possible TASs is large. Since the first effort in the MU–STLC systems is to reduce the complexity, an SINR-greedy-based TAS selection algorithm shown in Algorithm 1 has been proposed in [14].


**Algorithm 1** SINR-greedy-based TAS selection algorithm.
$$\begin{array}{l} \mathrm{Inputs}:\;H,Q_{2K}^{},N_{T},N_{S},K,\sigma_{x}^{2},\sigma_{z}^{2}\\ 1:\;C_{n}={\left\|\;H\;(:,n)\;\right\|}_{F}^{},\;n=1,2,\cdots ,N_{T}\\ 2:\;[\;V,\;u\;]=\max\left\{\;C_{1},C_{2},\cdots ,C_{N_{T}}\right\}\\ 3:\;H_{S}=H(\;:,u)\\ 4:\;{\bar{H}}_{0}=H(:,[1:(u-1)\;\;(u+1):\mathrm{end}])\\ 5:\;\mathrm{for}\;k=1,2,\cdots ,N_{S}-1\\ 6:\;\;\mathrm{for}\;q=1,2,\cdots ,(N_{T}-k)\\ 7:\;\;\;\;\;H_{temp}=\left[H_{S}(:,1:k)\;\;{\bar{H}}_{k-1}(:,q)\right]\\ 8:\;\;\;\;\;{\tilde{H}}_{temp}=\left[H_{temp}\;\;Q_{2K}^{T}H_{temp}^{*}\right]\\ 9:\;\;\;\;\;{\tilde{V}}_{k,q}={\left(\;{\tilde{H}}_{temp}^{}{\tilde{H}}_{temp}^{H}+\frac{2K\sigma_{z}^{2}}{\sigma_{x}^{2}}I_{2K}\right)}^{-1}\\ 10:\;\;\;\;\;\lambda_{k,q}=Tr\left[\;{\tilde{V}}_{k,q}\right]\\ 11:\;\;\mathrm{end}\\ 12:\;\;\hat{q}=\underset{q}{\;\arg\;\min}\;\lambda_{k,q}\\ 13:\;\;H_{S}(:,k+1)={\bar{H}}_{k-1}(:,\hat{q})\\ 14:\;\;{\bar{H}}_{k}={\bar{H}}_{k-1}(\;:,[1:(\hat{q}-1)\;(\hat{q}+1):\mathrm{end}])\\ 15:\;\mathrm{end}\\ \mathrm{Output}:\;H_{S} \end{array}$$



### 3.2. Proposed Incremental TAS Selection Algorithm

In the MMSE-precoded MU–STLC systems, the received SNR can be used as the design criterion for the proposed incremental TAS selection optimization and readily derived from (8), (10), and (12) as
(17)$$\eta_{S}={\tilde{\beta }}_{S}^{2}\frac{\sigma_{x}^{2}}{2\sigma_{z}^{2}}$$
where
(18)$${\tilde{\beta }}_{S}^{2}=\frac{1}{Tr\left[{\left(\;{\tilde{H}}_{S}\;{\tilde{H}}_{S}^{H}\;\right)}^{-1}\right]}$$

To significantly reduce the complexity of Algorithm 1, an SNR-based efficient incremental TAS selection algorithm can be developed by using the popular Woodbury formula [26]. The first-proposed TAS selection algorithm constructs the TAS by starting with an empty TAS and adding one antenna in each iteration. Assuming that k transmit antennas have been selected, the resulting selected subchannel is denoted as $H_{k}\in C^{2K\times \;\;k}$, where $1\leq k<N_{S}$. Then, the channel matrix for the (k+1)-th iteration process can be represented as
(19)$$H_{k+1}=\left[\;\;H_{k}h_{k+1}\right]$$
where $h_{k+1}$ denotes one of the unselected column vectors of H after completing the k-th iteration. Using the selected channel submatrix $H_{k}$, the (k+1)-th selected antenna can be determined by the following optimization.
(20)$$S_{k+1}=\underset{(k+1)\;\in R_{k}}{\arg\;\min}\;\;Tr\left[{\left(\;\;{\tilde{H}}_{k+1}{\tilde{H}}_{k+1}^{H}\right)}^{-1}\right]$$
where ${\tilde{H}}_{k+1}=\left[H_{k+1}\;\;Q_{2K}^{T}H_{k+1}^{*}\right]$, $S_{k+1}$ indicates the TAS determined after the (k+1)-th selection procedure, and $R_{k}$ denotes the TAS unselected after the k-th iteration. In each iteration, the computation of the matrix product ${\tilde{H}}_{k+1}{\tilde{H}}_{k+1}^{H}$ and the matrix inversion in ${\left(\;\;{\tilde{H}}_{k+1}{\tilde{H}}_{k+1}^{H}\right)}^{-1}$ requires an expensive computational load.

To reduce the computational complexity further, we adopt the Woodbury formula [26] written as
(21)$${\left(A+BC^{H}\right)}^{-1}=A^{-1}-A^{-1}B{\left(\;I+C^{H}A^{-1}B\right)}^{-1}C^{H}A^{-1}$$

Then, ${\left({\tilde{H}}_{k+1}{\tilde{H}}_{k+1}^{H}\right)}^{-1}$ can be re-expressed as
(22)$$\begin{array}{ll} {\left(\;{\tilde{H}}_{k+1}{\tilde{H}}_{k+1}^{H}\right)}^{-1} & ={\left(\;\;{\tilde{H}}_{k}{\tilde{H}}_{k}^{H}+{\tilde{h}}_{k+1}{\tilde{h}}_{k+1}^{H}\right)}^{-1}\\ & ={\left(\;\;{\tilde{H}}_{k}{\tilde{H}}_{k}^{H}\right)}^{-1}-{\left(\;\;{\tilde{H}}_{k}{\tilde{H}}_{k}^{H}\right)}^{-1}{\tilde{h}}_{k+1}{\left(\;I_{2}+{\tilde{h}}_{k+1}^{H}{\left(\;\;{\tilde{H}}_{k}{\tilde{H}}_{k}^{H}\right)}^{-1}{\tilde{h}}_{k+1}\right)}^{-1}{\tilde{h}}_{k+1}^{H}{\left(\;\;{\tilde{H}}_{k}{\tilde{H}}_{k}^{H}\right)}^{-1} \end{array}$$
where ${\tilde{h}}_{k+1}=\left[h_{k+1}\;\;Q_{2K}^{T}h_{k+1}^{*}\right]$. Thus, by defining ${\tilde{V}}_{k}={\left(\;\;{\tilde{H}}_{k}{\tilde{H}}_{k}^{H}\right)}^{-1}$, (20) can be written as
(23)$$S_{k+1}=\underset{(k+1)\in R_{k}}{\arg\;\min}\;\;Tr\left[{\tilde{V}}_{k}-{\tilde{V}}_{k}{\tilde{h}}_{k+1}{\left(\;I_{2}+{\tilde{h}}_{k+1}^{H}{\tilde{V}}_{k}{\tilde{h}}_{k+1}\right)}^{-1}{\tilde{h}}_{k+1}^{H}{\tilde{V}}_{k}\right]$$

Based on the above analysis, the procedure of the proposed SNR-based incremental TAS selection algorithm can be summarized in Algorithm 2. As an initial one of the matrix ${\tilde{V}}_{k}$, ${\tilde{V}}_{0}$ is employed with a 2K×2K identity matrix of $I_{2K}$. Here, H(:,q) and $H_{Q}(:,q)$ denote the *q*-th column vector of the updated channel matrix H and $H_{Q}=Q_{2K}^{T}H_{}^{*}$, respectively, which are associated with the transmit antennas remained after completing each step. It should be pointed out that in Algorithm 2, a 2×2 matrix inverse operation is performed in each incremental step, which mainly contributes to lower complexity compared to the conventional greedy-based algorithm. The idea of the proposed algorithm is to find a TAS by reducing the complexity for the matrix inverse operation needed at each step.


**Algorithm 2** SNR-based efficient incremental TAS selection algorithm.
$$\begin{array}{l} \mathrm{Inputs}:\;H,Q_{2K}^{},N_{T},N_{S},K\\ 1:\;H_{Q}=Q_{2K}^{T}H_{}^{*}\\ 2:\;{\tilde{V}}_{0}\;\;=I_{2K}\\ 3:\;\mathrm{for}\;k=1,2,\cdots ,N_{S}\\ 4:\;\;\mathrm{for}\;q=1,2,\cdots ,(N_{T}-k+1)\\ 5:\;\;\;\;\;{\tilde{H}}_{k,q}=\left[H(:,q)\;H_{Q}^{}(\;\;:\;\;,q)\right]\\ 6:\;\;\;\;\;{\tilde{\Theta }}_{k,q}={\tilde{V}}_{k-1}{\tilde{H}}_{k,q}\\ 7:\;\;\;\;\;\lambda_{k,q}=Tr\left[{\tilde{V}}_{k-1}-{\tilde{\Theta }}_{k,q}\;{\left(\;I_{2}+{\tilde{H}}_{k,q}^{H}{\tilde{\Theta }}_{k,q}\right)}^{-1}{\tilde{\Theta }}_{k,q}^{H}\right]\\ 8:\;\;\mathrm{end}\\ 9:\;\;\hat{q}=\underset{q}{\;\arg\;\min}\;\lambda_{k,q}\\ 10:\;\;{\tilde{V}}_{k}={\tilde{V}}_{k-1}-{\tilde{\Theta }}_{k,\hat{q}}\;{\left(\;I_{2}+{\tilde{H}}_{k,\hat{q}}^{H}{\tilde{\Theta }}_{k,\hat{q}}\right)}^{-1}{\tilde{\Theta }}_{k,\hat{q}}^{H}\\ 11:\;\;H=H(:,[1:(\hat{q}-1)\;\;(\hat{q}+1):\mathrm{end}])\\ 12:\;\;H_{Q}=H_{Q}(:,[1:(\hat{q}-1)\;\;(\hat{q}+1):\mathrm{end}])\\ 13:\;\mathrm{end}\\ 14:\;H_{S}=H\\ \mathrm{Output}:\;H_{S} \end{array}$$



### 3.3. Proposed Decremental TAS Selection Algorithm

In the ZF-precoded MU–STLC systems, the received SNR of (17) can be also employed as a design criterion for the proposed decremental TAS selection optimization. In order to see how TAS selection affects the SNR, let S and $S^{\prime }$ be two TASs in the ZF-precoded MU–STLC systems, where $S\subset S^{\prime }\subseteq \left\{\;1,\;2,\;\cdots \;,N_{T}\right\}$. Let $\bar{S}=S^{\prime }-S$ and $H_{S^{\prime }}=\left[\;H_{S}\;H_{\bar{S}}\right]$. Then, the expression of ${\tilde{H}}_{S^{\prime }}$ can be written as
(24)$${\tilde{H}}_{S^{\prime }}=\left[H_{S^{\prime }}\;\;Q_{2K}^{T}H_{S^{\prime }}^{*}\right]=\left[H_{S}\;H_{\bar{S}}\;\;\;Q_{2K}^{T}H_{S}^{*}\;\;\;Q_{2K}^{T}H_{\bar{S}}^{*}\right]$$

Thus, it is easily shown that ${\tilde{H}}_{S^{\prime }}^{}{\tilde{H}}_{S^{\prime }}^{H}={\tilde{H}}_{S}^{}{\tilde{H}}_{S}^{H}+{\tilde{H}}_{\bar{S}}^{}{\tilde{H}}_{\bar{S}}^{H}$, where ${\tilde{H}}_{\bar{S}}≜\left[H_{\bar{S}}\;\;Q_{2K}^{T}H_{\bar{S}}^{*}\right]$$\in C^{2K\times 2N_{S}}$. Now we have
(25)$${\left({\tilde{H}}_{S}^{}{\tilde{H}}_{S}^{H}\right)}^{-1}={\left({\tilde{H}}_{S^{\prime }}^{}{\tilde{H}}_{S^{\prime }}^{H}-{\tilde{H}}_{\bar{S}}^{}{\tilde{H}}_{\bar{S}}^{H}\right)}^{-1}$$

Then, according to the Woodbury formula [26] written as
(26)$${\left(A-BC^{H}\right)}^{-1}=A^{-1}+A^{-1}B{\left(I-C^{H}A^{-1}B\right)}^{-1}C^{H}A^{-1}$$
and the inverse of ${\tilde{H}}_{S}^{}{\tilde{H}}_{S}^{H}$ can be calculated as
(27)$${\left({\tilde{H}}_{S}^{}{\tilde{H}}_{S}^{H}\right)}^{-1}={\left({\tilde{H}}_{S^{\prime }}^{}{\tilde{H}}_{S^{\prime }}^{H}\right)}^{-1}+{\left({\tilde{H}}_{S^{\prime }}^{}{\tilde{H}}_{S^{\prime }}^{H}\right)}^{-1}{\tilde{H}}_{\bar{S}}^{}{\left(I-{\tilde{H}}_{\bar{S}}^{H}{\left({\tilde{H}}_{S^{\prime }}^{}{\tilde{H}}_{S^{\prime }}^{H}\right)}^{-1}{\tilde{H}}_{\bar{S}}^{}\right)}^{-1}{\tilde{H}}_{\bar{S}}^{H}{\left({\tilde{H}}_{S^{\prime }}^{}{\tilde{H}}_{S^{\prime }}^{H}\right)}^{-1}$$

Using (27), ${\tilde{\beta }}_{S}^{2}$ of (18) can be rewritten as
(28)$${\tilde{\beta }}_{S}^{2}=\frac{1}{Tr\left[{\left(\;\;{\tilde{H}}_{S^{\prime }}^{}{\tilde{H}}_{S^{\prime }}^{H}\right)}^{-1}\right]+Tr\left({\tilde{\Psi }}_{\bar{S}}\right)}$$
where
(29)$${\tilde{\Psi }}_{\bar{S}}={\tilde{V}}_{ZF,S^{\prime }}{\tilde{H}}_{\bar{S}}^{}{\left(I-{\tilde{H}}_{\bar{S}}^{H}{\tilde{V}}_{ZF,S^{\prime }}{\tilde{H}}_{\bar{S}}^{}\right)}^{-1}{\tilde{H}}_{\bar{S}}^{H}{\tilde{V}}_{ZF,S^{\prime }}$$
(30)$${\tilde{V}}_{ZF,S^{\prime }}={\left({\tilde{H}}_{S^{\prime }}^{}{\tilde{H}}_{S^{\prime }}^{H}\right)}^{-1}$$

Since ${\tilde{\Psi }}_{\bar{S}}>0$, we have
(31)$$P_{ZF,S}\;{\tilde{\beta }}_{S}^{2}={\tilde{\beta }}_{S^{\prime }}^{2}$$
where $P_{ZF,S}$ denotes the SNR penalty of TAS selection, which is defined as the increase in the received SNR for S to achieve the same SNR as that of $S^{\prime }$. The SNR penalty can be expressed as
(32)$$P_{ZF,S}≜\frac{{\tilde{\beta }}_{S^{\prime }}^{2}}{{\tilde{\beta }}_{S}^{2}}=\frac{Tr\left({\tilde{V}}_{S^{\prime }}\right)+Tr\left({\tilde{\Psi }}_{\bar{S}}\right)}{Tr\left({\tilde{V}}_{S^{\prime }}\right)}$$

If $Tr\left({\tilde{V}}_{S^{\prime }}\right)$ is assumed to be fixed, the SNR penalty can be minimized for the minimum value of $Tr\left({\tilde{\Psi }}_{\bar{S}}\right)$. Therefore, we obtain
(33)$$\begin{array}{ll} S_{opt} & =\underset{\begin{array}{c} S\in \left\{S_{n},n=1,2,\cdots ,C\left(N_{T},N_{S}\right)\right\}\\ \bar{S}=S^{\prime }-S \end{array}}{\arg\;\min}Tr\left({\tilde{\Psi }}_{\bar{S}}\right)\\ & =\underset{\begin{array}{c} S\in \left\{S_{n},n=1,2,\cdots ,C\left(N_{T},N_{S}\right)\right\}\\ \bar{S}=S^{\prime }-S \end{array}}{\arg\;\min}Tr\left[{\tilde{V}}_{S^{\prime }}{\tilde{H}}_{\bar{S}}^{}{\left(I-{\tilde{H}}_{\bar{S}}^{H}{\tilde{V}}_{S^{\prime }}{\tilde{H}}_{\bar{S}}^{}\right)}^{-1}{\tilde{H}}_{\bar{S}}^{H}{\tilde{V}}_{S^{\prime }}\right] \end{array}$$

The idea of the proposed TAS selection algorithms with lower complexity is to construct a TAS by removing one by one with a decremental manner from the full channel matrix H. The SNR-based decremental TAS selection algorithm for the ZF-precoded MU–STLC system is shown in Algorithm 3, which begins with a full channel matrix and eliminates one transmit antenna in each decremental step. Note that the initial matrix of ${\tilde{V}}_{S^{\prime }}$ is computed by ${\left({\tilde{H}}_{N_{T}}^{}{\tilde{H}}_{N_{T}}^{H}\right)}^{-1}$, where ${\tilde{H}}_{N_{T}}=\left[H\;Q_{2K}^{T}H_{}^{*}\right]$. The matrix dimension of ${\tilde{H}}_{N_{T}-k+1}^{}$ used in computing ${\tilde{V}}_{N_{T}-k+1}$ becomes smaller at each iteration.


**Algorithm 3**
SNR-based decremental TAS selection algorithm.
$$\begin{array}{l} \mathrm{Inputs}:\;H,Q_{2K}^{},N_{T},N_{S}\\ 1:\;H_{S^{\prime }}=H\;\\ 2:\;\mathrm{for}\;k=1,2,\cdots ,N_{T}-N_{S}\\ 3:\;\;\;H_{Q}=Q_{2K}^{T}H_{S^{\prime }}^{*}\\ 4:\;\;\;{\tilde{H}}_{N_{T}-k+1}=\left[H_{S^{\prime }}\;\;H_{Q}\right]\\ 5:\;\;\;{\tilde{V}}_{N_{T}-k+1}={\left({\tilde{H}}_{N_{T}-k+1}^{}{\tilde{H}}_{N_{T}-k+1}^{H}\right)}^{-1}\\ 6:\;\;\;\mathrm{for}\;q=1,2,\cdots ,(N_{T}-k+1)\\ 7:\;\;\;\;{\tilde{H}}_{\bar{S}}=\left[H_{S^{\prime }}(:,q)\;H_{Q}^{}(:,q)\right]\\ 8:\;\;\;\;{\tilde{\Theta }}_{k,q}={\tilde{V}}_{N_{T}-k+1}{\tilde{H}}_{\bar{S}}^{}\\ 9:\;\;\;\;\lambda_{k,q}=Tr\left[{\tilde{\Theta }}_{k,q}{\left(I_{2}-{\tilde{H}}_{\bar{S}}^{H}{\tilde{\Theta }}_{k,q}\right)}^{-1}{\tilde{\Theta }}_{k,q}^{H}\right]\\ 10:\;\;\;\mathrm{end}\\ 11:\;\;\;\hat{q}=\underset{q}{\;\arg\;\min}\;\lambda_{k,q}\\ 12:\;\;\;H_{S^{\prime }}=H_{S^{\prime }}(:,[1:(\hat{q}-1)\;(\hat{q}+1):\mathrm{end}])\\ 13:\;\mathrm{end}\\ 14:\;H_{S}=H_{S^{\prime }}\\ \mathrm{Output}:\;H_{S} \end{array}$$



To reduce the complexity of Algorithm 3 further, a ZF-based efficient decremental TAS selection algorithm summarized in Algorithm 4 can be developed by using (27). The first step starts with a full channel matrix and removes one transmit antenna in each decremental step. After taking k decremental steps, k transmit antennas are removed, and then the corresponding remained channel submatrix is denoted by $H_{N_{T}-k}\in C^{2K\times \;\;(N_{T}-k)}$, where $1\leq k\leq (N_{T}-N_{S})$, and can be expressed as
(34)$$H_{N_{T}-k}=\left[\;H_{N_{T}-k-1}h_{k}\right]$$
where $h_{k}$ is the column vector of $H_{N_{T}-k}$ corresponding to the (k+1)-th deleted antenna. Then, based on (33), the proposed ZF-based efficient decremental TAS selection problem can be rewritten as
(35)$$S_{k}=\underset{S_{k}}{\arg\;\min}\;Tr\left[{\tilde{V}}_{N_{T}-k}{\tilde{H}}_{k}^{}{\left(I_{2}-{\tilde{H}}_{k}^{H}{\tilde{V}}_{N_{T}-k}{\tilde{H}}_{k}^{}\right)}^{-1}{\tilde{H}}_{k}^{H}{\tilde{V}}_{N_{T}-k}\right]$$
where $S_{k}$ denotes the antenna subset remained at the k-th decremental step and
(36)$${\tilde{V}}_{N_{T}-k}={\left({\tilde{H}}_{N_{T}-k}^{}{\tilde{H}}_{N_{T}-k}^{H}\right)}^{-1}$$
(37)$${\tilde{H}}_{N_{T}-k}≜\left[H_{N_{T}-k}\;\;Q_{2K}^{T}H_{N_{T}-k}^{*}\right]=\left[H_{N_{T}-k-1}\;h_{k}\;\;\;Q_{2K}^{T}H_{N_{T}-k-1}^{*}\;\;\;Q_{2K}^{T}h_{k}^{*}\right]$$
(38)$${\tilde{H}}_{k}≜\left[h_{k}\;\;Q_{2K}^{T}h_{k}^{*}\right]\in C^{2K\times 2}$$

In Algorithm 4, the computation of ${\tilde{V}}_{N_{T}}$ using matrix inverse operation is carried out only once in the beginning, and then the matrix ${\tilde{V}}_{N_{T}-k+2}$ is updated using ${\tilde{V}}_{N_{T}-k+1}$ obtained at the previous iteration. This procedure is different from Algorithm 3 and thus makes a contribution to have lower complexity.


**Algorithm 4** Proposed ZF-based efficient decremental TAS selection algorithm.
$$\begin{array}{l} \mathrm{Inputs}:\;H,Q_{2K}^{},N_{T},N_{S}\\ 1:\;H_{Q}=Q_{2K}^{T}H_{}^{*}\\ 2:\;{\tilde{H}}_{S^{\prime }}=\left[H\;\;H_{Q}\right]\\ 3:\;{\tilde{V}}_{N_{T}}={\left(\;{\tilde{H}}_{S^{\prime }}^{}{\tilde{H}}_{S^{\prime }}^{H}\right)}^{-1}\\ 4:\;\mathrm{for}\;k=1,2,\cdots ,N_{T}-N_{S}\\ 5:\;\;\mathrm{for}\;q=1,2,\cdots ,(N_{T}-k+1)\\ 6:\;\;\;\;\;{\tilde{H}}_{k,q}=\left[H(:,q)\;\;H_{Q}^{}(:,q)\right]\\ 7:\;\;\;\;\;{\tilde{\Theta }}_{k,q}={\tilde{V}}_{N_{T}-k+1}{\tilde{H}}_{k,q}\\ 8:\;\;\;\;\;\lambda_{k,q}=Tr\left[{\tilde{\Theta }}_{k,q}\;{\left(\;I_{2}-{\tilde{H}}_{k,q}^{H}{\tilde{\Theta }}_{k,q}\right)}^{-1}{\tilde{\Theta }}_{k,q}^{H}\right]\\ 9:\;\;\mathrm{end}\\ 10:\;\;\hat{q}=\underset{q}{\;\arg\;\min}\;\lambda_{k,q}\\ 11:\;\;{\tilde{V}}_{N_{T}-k+2}={\tilde{V}}_{N_{T}-k+1}+{\tilde{\Theta }}_{k,\hat{q}}\;\;{\left(\;I_{2}-{\tilde{H}}_{k,\hat{q}}^{H}{\tilde{\Theta }}_{k,\hat{q}}\right)}^{-1}{\tilde{\Theta }}_{k,\hat{q}}^{H}\\ 12:\;\;H=H(:,[1:(\hat{q}-1)(\hat{q}+1):\mathrm{end}])\\ 13:\;\;H_{Q}=H_{Q}(:,[1:(\hat{q}-1)(\hat{q}+1):\mathrm{end}])\\ 14:\;\mathrm{end}\\ 15:\;H_{S}=H\\ \mathrm{Output}:\;H_{S} \end{array}$$



In the MMSE-precoded MU–STLC systems, the mean square error (MSE) criterion can be adopted for the proposed efficient decremental TAS selection. For a channel matrix $H_{S}$, the MSE derived in [14] for MMSE-precoded MU–STLC systems is given as
(39)$$J_{S}=2K\sigma_{z}^{2}\;Tr\left(\;{\tilde{V}}_{S}\right)$$


Similar to the ZF-precoded MU–STLC systems, we want to see how TAS selection affects the MSE. By denoting S and $S^{\prime }$ by two TASs to be satisfied with $S\subset S^{\prime }\subseteq \left\{\;1,\;2,\;\cdots \;,N_{T}\right\}$ and $\bar{S}=S^{\prime }-S$, the MSE difference between two sets of $S^{\prime }$ and S in MMSE-precoded MU–STLC systems can be written as
(40)$$\begin{array}{ll} J_{\bar{S}} & =J_{S^{\prime }}-J_{S}=2K\sigma_{z}^{2}Tr\left({\tilde{V}}_{S^{\prime }}-{\tilde{V}}_{S}\right)\\ & =2K\sigma_{z}^{2}\;Tr\left(\;{\tilde{V}}_{S^{\prime }}-{\left({\tilde{H}}_{S^{\prime }}^{}{\tilde{H}}_{S^{\prime }}^{H}-{\tilde{H}}_{\bar{S}}^{}{\tilde{H}}_{\bar{S}}^{H}\right)}^{-1}\right)\\ & =-2K\sigma_{z}^{2}\;Tr\left(\;{\tilde{V}}_{S^{\prime }}{\tilde{H}}_{\bar{S}}^{}{\left(\;I_{2K}-{\tilde{H}}_{\bar{S}}^{H}{\tilde{V}}_{S^{\prime }}{\tilde{H}}_{\bar{S}}^{}\right)}^{-1}{\tilde{H}}_{\bar{S}}^{H}{\tilde{V}}_{S^{\prime }}\right) \end{array}$$
where (41)$${\tilde{V}}_{S^{\prime }}={\left({\tilde{H}}_{S^{\prime }}^{}{\tilde{H}}_{S^{\prime }}^{H}+\frac{2K\sigma_{z}^{2}}{\sigma_{x}^{2}}I_{2K}\right)}^{-1}$$
(42)$${\tilde{V}}_{\bar{S}}={\left({\tilde{H}}_{\bar{S}}^{}{\tilde{H}}_{\bar{S}}^{H}+\frac{2K\sigma_{z}^{2}}{\sigma_{x}^{2}}I_{2K}\right)}^{-1}$$
(43)$${\tilde{H}}_{S^{\prime }}≜\left[H_{S^{\prime }}\;\;Q_{2K}^{T}H_{S^{\prime }}^{*}\right]$$
(44)$${\tilde{H}}_{\bar{S}}≜\left[H_{\bar{S}}\;\;Q_{2K}^{T}H_{\bar{S}}^{*}\right]$$

Since $Tr\left(\;{\tilde{V}}_{S^{\prime }}{\tilde{H}}_{\bar{S}}^{}{\left(\;I_{2K}-{\tilde{H}}_{\bar{S}}^{H}{\tilde{V}}_{S^{\prime }}{\tilde{H}}_{\bar{S}}^{}\right)}^{-1}{\tilde{H}}_{\bar{S}}^{H}{\tilde{V}}_{S^{\prime }}\right)>0$, $J_{\bar{S}}$ of (40) becomes negative. Thus, it can be concluded that when the transmit power constraint and the number $N_{T}$ of transmit antennas are fixed, the MSE is monotonically decreasing with the number $N_{S}$ of active transmit antennas in MMSE-precoded MU–STLC systems.

Then, the optimal TAS selection algorithm based on the MSE criterion for the MMSE-precoded MU–STLC system can be formulated as [14].

(45)$$\begin{array}{l} S_{opt}=\underset{\begin{array}{l} S\in \;\left\{\;S_{n},n=1,2,\cdots ,C\;\left(N_{T},N_{S}\right)\;\right\}\\ \;\bar{S}=S^{\prime }-S \end{array}}{arg min}\;\left|\;J_{\bar{S}}\;\right|\\ \;=\underset{\begin{array}{l} S\in \;\left\{S_{n},n=1,2,\cdots ,C\;\left(N_{T},N_{S}\right)\right\}\\ \;\bar{S}=S^{\prime }-S \end{array}}{arg min}Tr\left[{\tilde{V}}_{S^{\prime }}{\tilde{H}}_{\bar{S}}^{}{\left(\;I_{2K}-{\tilde{H}}_{\bar{S}}^{H}{\tilde{V}}_{S^{\prime }}{\tilde{H}}_{\bar{S}}^{}\right)}^{-1}{\tilde{H}}_{\bar{S}}^{H}{\tilde{V}}_{S^{\prime }}\right] \end{array}$$
which is the same as that for the ZF-precoded MU–STLC system, except for ${\tilde{V}}_{S^{\prime }}$ given as (41). Therefore, for MSE-based efficient decremental TAS selection in MMSE-precoded MU–STLC systems, line 3 in Algorithm 4 is replaced with the following:(46)$${\tilde{V}}_{N_{T}}\;\;={\left(\;{\tilde{H}}_{S^{\prime }}^{}{\tilde{H}}_{S^{\prime }}^{H}+\frac{2K\sigma_{z}^{2}}{\sigma_{x}^{2}}I_{2K}\right)}^{-1}$$

On the other hand, the received SINR penalty in MMSE-precoded MU–STLC systems can be defined as
(47)$$P_{MMSE,S}≜\frac{\gamma_{S^{\prime }}}{\gamma_{S}}$$
where the received SINRs for two TASs of $S^{\prime }$ and S in MMSE-precoded MU–STLC systems are given as [14].
(48)$$\gamma_{S^{\prime }}=\frac{\sigma_{x}^{2}}{\sigma_{z}^{2}\;Tr\left(\;{\tilde{V}}_{S^{\prime }}\right)}-1$$
(49)$$\gamma_{S}=\frac{\sigma_{x}^{2}}{\sigma_{z}^{2}\;Tr\left(\;{\tilde{V}}_{S}\right)}-1$$

The asymptotic received SINR penalty for large values of SNR in the MMSE- precoded MU–STLC systems can be obtained as
(50)$$\begin{array}{ll} \underset{SNR\to \infty }{\lim}P_{MMSE,S} & =\underset{SNR\to \infty }{\lim}\frac{\frac{1}{Tr\left(\;{\tilde{V}}_{MMSE,S^{\prime }}\right)}-\frac{1}{SNR}}{\frac{1}{Tr\left(\;{\tilde{V}}_{MMSE,S}\right)}-\frac{1}{SNR}}=\underset{SNR\to \infty }{\lim}\frac{Tr\left(\;{\tilde{V}}_{MMSE,S}\right)}{Tr\left(\;{\tilde{V}}_{MMSE,S^{\prime }}\right)}\\ & =\underset{SNR\to \infty }{\lim}\frac{Tr\left(\;{\left(\;{\tilde{H}}_{S}^{}{\tilde{H}}_{S}^{H}+\frac{2K}{SNR}I_{2K}\right)}^{-1}\right)}{Tr\left(\;{\left(\;{\tilde{H}}_{S^{\prime }}^{}{\tilde{H}}_{S^{\prime }}^{H}+\frac{2K}{SNR}I_{2K}\right)}^{-1}\right)}=\frac{Tr\left(\;{\tilde{V}}_{ZF,S}\right)}{Tr\left(\;{\tilde{V}}_{ZF,S^{\prime }}\right)}=\frac{Tr\left(\;{\tilde{V}}_{S^{\prime }}\right)+Tr\left(\;{\tilde{\Psi }}_{\bar{S}}\right)}{Tr\left(\;{\tilde{V}}_{S^{\prime }}\right)} \end{array}$$
where $\mathrm{SNR}=\sigma_{x}^{2}/\sigma_{z}^{2}$. The asymptotic received SINR penalty of (50) has the same expression as that of (32) in the ZF-precoded MU–STLC systems.

## 4. Complexity Analysis

For complexity analysis of TAS selection algorithms, we consider the number of real multiplications (RMs) and the number of real summations (RSs) [24,27]. Given arbitrary matrices $A\in C^{N\times M}$ and $B\in C^{M\times P}$, the number of complex multiplications (CMs) and complex summations (CSs) required for three matrix-related operations is given in Table 1 [27], which is utilized in the following complexity analysis. Here, a CM requires four RMs and two RSs, whereas a CS uses two RSs.

Recall that this work has employed an $N_{T}-\mathrm{by}-2K$ MU–STLC system with $N_{T}$ available transmit antennas, $N_{S}$ selected transmit antennas, and K users, where the receiver of each user has two receive antennas for STLC. Thus, it is assumed that the total number of receive antennas is 2K.

### 4.1. Complexity of SINR-Greedy-Based TAS Selection Algorithm

From Algorithm 1, the computational complexities of the SINR-greedy-based TAS selection scheme in terms of RMs and RSs, respectively, can be evaluated line by line as

Line 1:RM in $C_{n}={\left\|H(:,n)\right\|}_{F}^{}\Rightarrow 8KN_{T}$,RS in $C_{n}={\left\|H(:,n)\right\|}_{F}^{}\Rightarrow 8KN_{T}-2N_{T}^{}$,

Line 9:RM in $\tilde{\Phi }={\tilde{H}}_{temp}^{}{\tilde{H}}_{temp}^{H}+(2K\sigma_{z}^{2}/\sigma_{x}^{2})I_{2K}\Rightarrow (16K^{2}+8K)(k+1)+3$,RS in $\tilde{\Phi }={\tilde{H}}_{temp}^{}{\tilde{H}}_{temp}^{H}+(2K\sigma_{z}^{2}/\sigma_{x}^{2})I_{2K}\Rightarrow (4K^{2}+2K)(4k+3)+2K$,RM in ${\tilde{\Phi }}^{-1}\Rightarrow 16K^{3}+24K^{2}$,RS in ${\tilde{\Phi }}^{-1}\Rightarrow 16K^{3}+8K^{2}$,

Line 10:RS in $Tr[\;{\tilde{V}}_{k,q}]\Rightarrow 4K$,

Thus, the overall complexities of the SINR-greedy-based TAS selection algorithm (called greedy) in terms of RMs and RSs, respectively, are given as
(51)$$\begin{array}{l} N_{greedy}^{RM}=K^{3}\left(16N_{T}N_{S}+8N_{S}^{}-16N_{T}-8N_{S}^{2}\right)+K^{2}\left(8N_{T}N_{S}^{2}+32N_{T}N_{S}+\frac{52}{3}N_{S}-40N_{T}-12N_{S}^{2}-\frac{16}{3}N_{S}^{3}\right)+K\left(4N_{T}N_{S}^{2}+4N_{T}N_{S}^{}+\frac{8}{3}N_{S}-\frac{8}{3}N_{S}^{3}\right)+3N_{T}N_{S}+\frac{3}{2}N_{S}^{}-3N_{T}-\frac{3}{2}N_{S}^{2} \end{array}$$
(52)$$\begin{array}{l} N_{greedy}^{RS}=K^{3}\left(16N_{T}N_{S}+8N_{S}^{}-16N_{T}-8N_{S}^{2}\right)+K^{2}\left(8N_{T}N_{S}^{2}+12N_{T}N_{S}+\frac{22}{3}N_{S}^{}-20N_{T}-2N_{S}^{2}-\frac{16}{3}N_{S}^{3}\right)+K\left(4N_{T}N_{S}^{2}+8N_{T}N_{S}+\frac{14}{3}N_{S}^{}-4N_{T}-2N_{S}^{2}-\frac{8}{3}N_{S}^{3}\right)-2N_{T} \end{array}$$

### 4.2. Complexity of Proposed SNR-Based Efficient Incremental TAS Selection Algorithm

From Algorithm 2 with (23) in lines 6 and 7, the numbers of RMs and RSs for the proposed SNR-based efficient incremental TAS selection algorithm can be calculated as

Line 6:RM in ${\tilde{V}}_{k-1}{\tilde{H}}_{k,q}\Rightarrow 32K^{2}$,RS in ${\tilde{V}}_{k-1}{\tilde{H}}_{k,q}\Rightarrow 32K^{2}-8K$,

Line 7:RM in $\mu ={\tilde{H}}_{k,q}^{H}{\tilde{\Theta }}_{k,q}\Rightarrow 32K$,RS in $\mu ={\tilde{H}}_{k,q}^{H}{\tilde{\Theta }}_{k,q}\Rightarrow 24K-6$,RM in $\Delta ={\left(I_{2}-\mu \right)}^{-1}\Rightarrow 40$,RS in $\Delta ={\left(I_{2}-\mu \right)}^{-1}\Rightarrow 26$,RM in $\alpha =\Delta {\tilde{\Theta }}_{k,q}^{H}\Rightarrow 32K$RS in $\alpha =\Delta {\tilde{\Theta }}_{k,q}^{H}\Rightarrow 24K$,RM in $\psi ={\tilde{\Theta }}_{k,q}\alpha \Rightarrow 16K^{2}+8K$,RS in $\psi ={\tilde{\Theta }}_{k,q}\alpha \Rightarrow 28K^{2}+14K$RS in ${\tilde{V}}_{k-1}-\psi \Rightarrow 4K$,RS in Tr[ψ]⇒4K,

Thus, the overall computational complexities of the proposed SNR-based incremental TAS selection algorithm (called proposed incremental) are given by
(53)$$N_{proposed-incremental}^{RM}=\left(24K^{2}+38K+20\right)\left(2N_{T}N_{S}+N_{S}-N_{S}^{2}\right)$$
(54)$$N_{proposed-incremental}^{RS}=\left(30K^{2}+29K+10\right)\left(2N_{T}N_{S}+N_{S}-N_{S}^{2}\right)$$

### 4.3. Complexity of Proposed MMSE-Based Decremental TAS Selection Algorithm

From Algorithm 4 with (46) in line 3, the numbers of RMs and RSs for the proposed MMSE-based decremental TAS selection algorithm can be obtained as

Line 3: RMs and RSs can be computed in a similar manner used in line 9 of Section 4.1.

Line 7:RM in ${\tilde{\Theta }}_{k,q}={\tilde{V}}_{N_{T}-k+1}{\tilde{H}}_{k,q}\Rightarrow 32K^{2}$,RS in ${\tilde{\Theta }}_{k,q}={\tilde{V}}_{N_{T}-k+1}{\tilde{H}}_{k,q}\Rightarrow 32K^{2}-8K$,

Line 8: RMs and RSs are the same as those, except for RS in ${\tilde{V}}_{k-1}-\psi$, in line 7 of Section 4.2.

Line 11: RSs are evaluated as in line 7 of Section 4.1.

Thus, the overall complexities of the proposed MMSE-based decremental TAS selection algorithm (called proposed MMSE incremental) in terms of RMs and RSs, respectively, are given as
(55)$$\begin{array}{l} N_{proposed-MMSE-decremental}^{RM}=16K^{3}+K^{2}\left(24N_{T}^{2}+40N_{T}+24-24N_{S}^{2}-24N_{S}\right)+K\left(36N_{T}^{2}+44N_{T}-36N_{S}^{2}-36N_{S}\right)+20\left(N_{T}^{2}+N_{T}-N_{S}^{2}-N_{S}\right) \end{array}$$
(56)$$\begin{array}{l} N_{proposed-MMSE-decremental}^{RS}=16K^{3}+K^{2}\left(30N_{T}^{2}+46N_{T}+4-30N_{S}^{2}-30N_{S}\right)+K\left(27N_{T}^{2}+39N_{T}-27N_{S}^{2}-31N_{S}-2\right)+10\left(N_{T}^{2}+N_{T}-N_{S}^{2}-N_{S}\right) \end{array}$$

### 4.4. Complexity Comparison

In this work, $\left(N_{T},N_{S},K\right)$ indicates that $N_{T}$ transmit antennas, $N_{S}$ selected transmit antennas, and K users are used as system parameters. Remember that each user has two receive antennas, even for the MU–STLC systems. In Figure 1, the complexity of the proposed incremental and decremental TAS selection algorithms is compared with that of the greedy algorithm as a function of the number of selected transmit antennas for $N_{T}=8$, where the number of users is assumed to be the same as that of the selected transmit antennas. It is deduced that for the small number of $N_{S}(=2,3)$ in the $\left(8,N_{S},K\right)$ MU–STLC system with $N_{S}=K$, the complexity of the conventional greedy-based algorithm is comparable to the proposed algorithms. As $N_{S}$ gets higher, the complexity of the greedy algorithm significantly increases and the proposed incremental algorithm obtains a much smaller complexity than the greedy one. On the other hand, the decremental algorithm achieves the smallest complexity for large $N_{S}$.

Figure 2a–d illustrate the complexity of three TAS selection algorithms as a function of varying $N_{S}$ for $\left(8,N_{S},4\right)$, $\left(16,N_{S},4\right)$, $\left(32,N_{S},8\right)$ and $\left(200,N_{S},32\right)$, and MU–STLC systems, respectively, where the number of users is invariant. For the small number of $N_{S}(=4,5)$ of the $\left(16,N_{S},4\right)$ MU–STLC system in Figure 2b, the conventional greedy-based algorithm has lower complexity than the proposed MMSE decremental algorithm but higher complexity than the proposed incremental algorithm. Meanwhile, it can be observed that the proposed incremental algorithm significantly reduces the complexity of the greedy algorithm for larger $N_{S}$. In addition, the rate of increase for the proposed incremental algorithm is much smaller than that of the greedy algorithm. On the other hand, the complexity of the decremental algorithm becomes lower as $N_{S}$ increases. It is mainly due to reduction of $C(N_{T},N_{S})$. Furthermore, it is found that the decremental algorithm can attain lower complexity than the proposed incremental algorithm when is greater than half of $N_{T}$. It should be noted that the proposed algorithms conduct 2×2 matrix inverse operations in each incremental/decremental step, which is the main reason to offer a low complexity for large antenna dimensions, compared to the conventional greedy-based algorithm requiring 2K×2K matrix inverse operations.

## 5. Simulation Results

In this section, several TAS selection algorithms for the MU–STLC system with $N_{T}$ transmit antennas and K users are evaluated through Monte Carlo simulations over static Raleigh flat-fading channels. Each user has two receive antennas. The SNR is defined as $\sigma_{x}^{2}/\sigma_{z}^{2}$. In the case of $N_{S}=N_{T}$, there is no TAS selection. We assume that the CSI is perfectly known at the transmitter of the MU–STLC system. The quadrature phase shift keying (QPSK) modulation is assumed. In the simulations, the BER performance of the linear precoded MU–STLC system is compared using the following TAS selection algorithms:(a)SINR-greedy-based TAS selection [14] (greedy);(b)Proposed SNR-based efficient incremental TAS selection (proposed incremental);(c)Proposed ZF-based efficient decremental TAS selection (proposed ZF decremental);(d)Proposed MMSE-based efficient decremental TAS selection (proposed MMSE decremental);(e)Optimal ZF-based TAS selection (optimal ZF);(f)Optimal MMSE-based TAS selection (optimal MMSE).

It should be pointed out that the linear precoded MU–STLC TAS systems with (a), (b), (d), and (f) employ MMSE precoders at the transmitter, whereas the systems based on TAS selection of (c) and (e) use ZF precoders.

The BER performance of the proposed TAS selection algorithms is given in Figure 3 in the scenario of $\left(N_{T},N_{S},K)=(8,4,4\right)$. We observe that the proposed incremental TAS selection algorithm achieves a slightly worse performance than the greedy algorithm in the high SNR regime with lower complexity (which is confirmed from Figure 1). It is also shown that the proposed decremental TAS selection algorithm outperforms the greedy and proposed incremental algorithms. The performance of the latter may be poorly affected by an initially selected antenna. Furthermore, the proposed MMSE decremental TAS selection algorithm can achieve better BER performance compared to that of the proposed ZF decremental TAS selection one. In addition, the BER results of the proposed ZF and MMSE decremental selection algorithms are close to those of the optimal exhaustive search-based ZF and MMSE TAS selection algorithms, respectively.

Figure 4 depicts the BER results of the proposed TAS selection algorithms for the $\left(8,N_{S},K\right)$ MU–STLC system when $N_{S}$ is equal to K. It is assumed that the values of K are given as K=2,4, and 8. Note that (8,8,8) indicates no TAS selection case. It is observed that the BER performance of all the TAS selection algorithms improves as K decreases. The good performance for small K results from the decreased multiuser interference. Especially, the proposed MMSE decremental TAS selection algorithm is shown to provide the best BER performance. In this scenario, the computational complexity of the greedy, the proposed incremental, and the proposed MMSE decremental algorithms is shown in Figure 1. For $N_{S}=2$, the complexity of the greedy algorithm is the smallest, while the BER performance of the greedy algorithm is slightly less than that of the proposed MMSE decremental algorithm and similar to that of the proposed incremental algorithm. We can observe that the BER performance difference between the proposed ZF and MMSE decremental algorithms becomes smaller as the diversity gain gets larger. It should be noted that the diversity gain of the (8,2,2) MU–STLC system is larger than that of the (8,4,4) case. 

In Figure 5 and Figure 6, as the number of active transmit antennas decreases in the proposed MMSE decremental TAS selection algorithm, the received SNR penalty increases. Here, the number of users is fixed as K=4. It is also seen in Figure 5 that the performance gap between the proposed decremental selection algorithms and the greedy algorithm is bigger as the number of selected antennas is smaller. It is observed in Figure 6 that when the diversity gain is high, all the greedy, the proposed incremental, and the proposed MMSE decremental algorithms offer similar BER performance. In Figure 5, the SNR penalties are approximately given as 1 dB and 3.8 dB, respectively, for $N_{S}=6$ and $N_{S}=4$. In Figure 6, they are approximated by 2.5 dB and 6.75 dB, respectively, for $N_{S}=8$ and $N_{S}=4$. They are well agreed with analytical results in Figure 7, which is obtained from (50). Thus, we can evaluate the received SNR penalty owing to TAS selection by using (50) for massive MU–STLC MIMO systems. Figure 8 exhibits the asymptotic received SINR penalty as a function of $N_{S}$ for the MU–STLC system with $N_{T}=32$ and K=8. It is found that as the number of selected transmit antennas increases, the SINR penalty decreases. Furthermore, we can expect from the received SINR penalty analysis that without BER simulations, the BER results of the proposed MMSE decremental and conventional greedy algorithms are similar for the $\left(32,N_{S},8\right)$ MU–STLC system.

Finally, Figure 9 and Figure 10 illustrate the BER results of the proposed TAS selection algorithms as a function of K for the $\left(8,N_{S},K\right)$ MU–STLC system with $N_{S}=K$ when SNR=12 dB and SNR=16 dB, respectively, are given. The BER performance worsens for all TAS selection algorithms as $N_{S}(=K)$ increases. It is due to the increased multiuser interference. Particularly, the proposed ZF decremental TAS selection algorithm for large K achieves poor performance owing to large multiuser interference. On the other hand, it is seen that the proposed MMSE decremental TAS selection algorithm achieves the BER performance close to the optimal algorithm and outperforms the other TAS selection schemes for $N_{S}<N_{T}$. This implies that the proposed MMSE decremental selection algorithm is able to effectively suppress the multiuser interference. For this scenario, the comparison of computational complexity has been given in Figure 1. Recall that the proposed MMSE decremental TAS selection algorithm has the lowest complexity for $N_{S}=\{5,6,7\}$.

## 6. Conclusions

This paper proposes incremental and decremental TAS selection algorithms with low complexity to efficiently reduce the number of RF chains for the MU–STLC systems. It is analyzed that the proposed algorithms achieve significantly reduced complexity than the existing greedy algorithm for the MU–STLC systems. Thus, they achieve a better tradeoff between BER performance and computational complexity for large antenna dimensions. The proposed MMSE decremental TAS selection algorithm obtains better BER performance than the existing greedy algorithm for the MU–STLC systems with relatively low antenna dimensions. Its BER performance is close to that achieved by exhaustive search-based optimal MMSE-based TAS selection algorithm for the (8,4,4) MU–STLC system. Its complexity is significantly lower than the other algorithms especially for large $N_{S}$. Moreover, it is analytically shown that decreasing the number of active transmit antennas in the MU–STLC systems damages the detection SINR. Its simulation results are shown to agree with the asymptotic received SINR penalty. As possible future research, it would be worthwhile to consider the joint transmit/receive antenna subsets selection or joint transmit antenna/user subset selection for the MU–STLC systems.

## Figures and Tables

**Figure 1 sensors-21-02690-f001:**
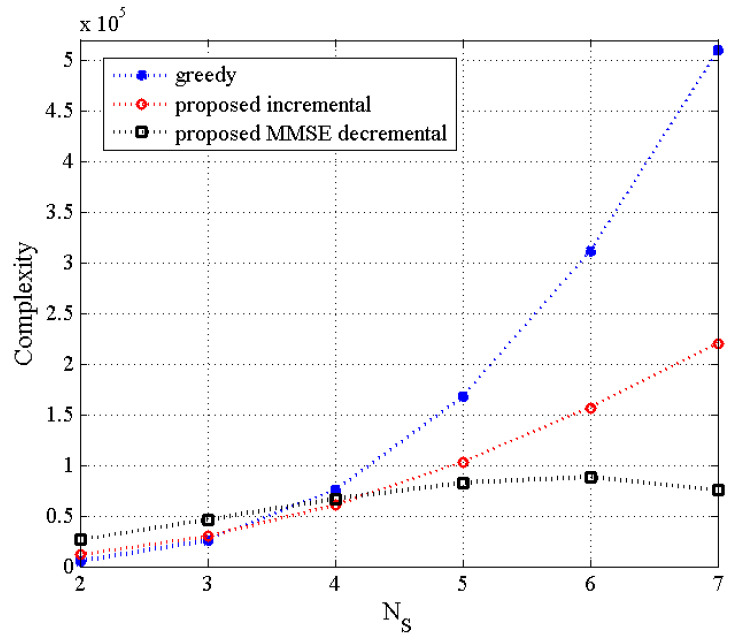
The complexity of the proposed transmit antenna subset (TAS) selection algorithms as a function of $N_{S}$ for $\left(8,N_{S},K\right)$ multiuser space–time line code (MU–STLC) system with $N_{S}=K$.

**Figure 2 sensors-21-02690-f002:**
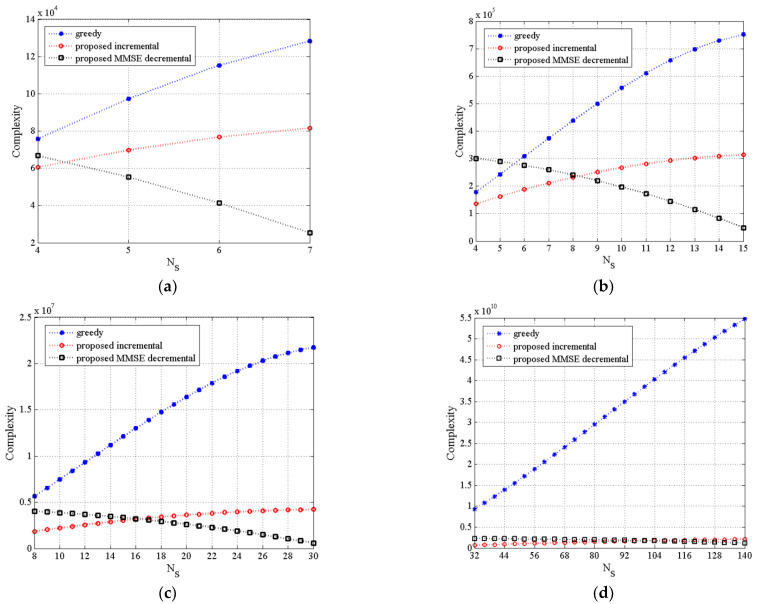
The complexity of the proposed TAS selection algorithms as a function of $N_{S}$ for (**a**)$\left(8,N_{S},4\right)$, (**b**)$\left(16,N_{S},4\right)$, (**c**)$\left(32,N_{S},8\right)$, and (**d**)$\left(200,N_{S},32\right)$ MU–STLC systems.

**Figure 3 sensors-21-02690-f003:**
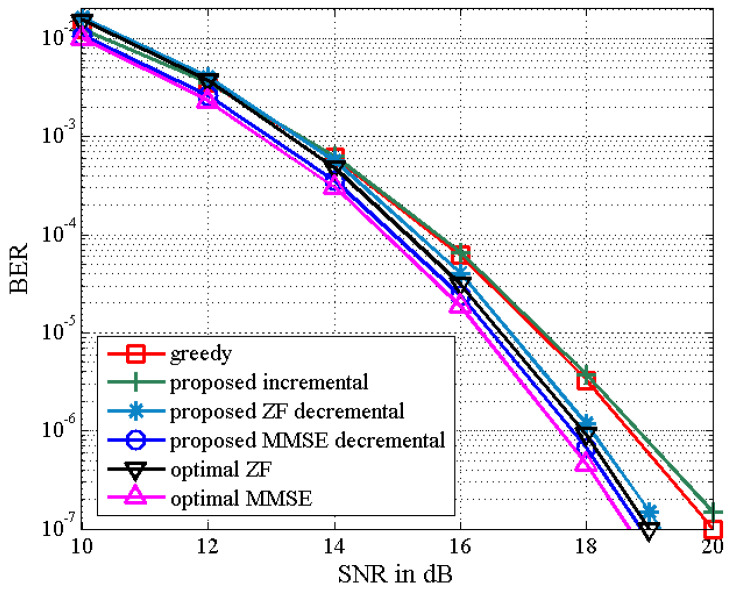
Bit error rate (BER) of the proposed TAS selection algorithm and optimal selection algorithm for (8,4,4) MU–STLC system.

**Figure 4 sensors-21-02690-f004:**
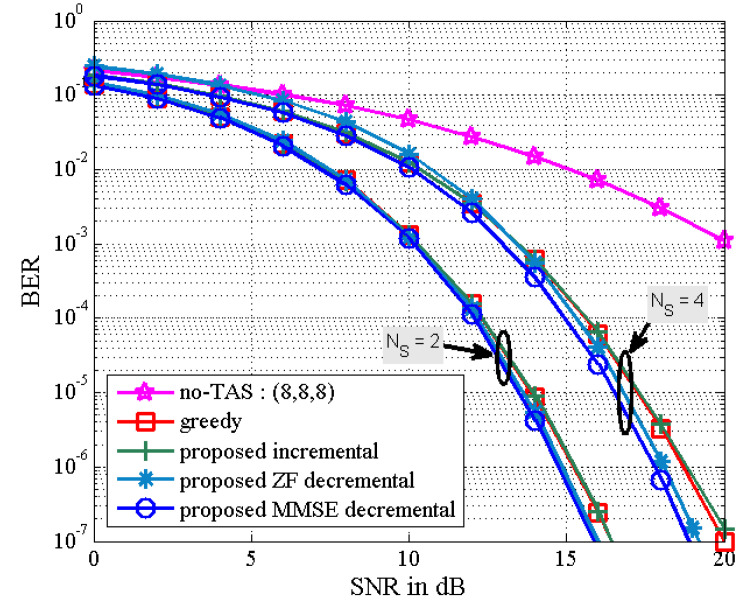
BER the proposed TAS selection algorithms for $\left(8,N_{S},K\right)$ MU–STLC system with $N_{S}=K$.

**Figure 5 sensors-21-02690-f005:**
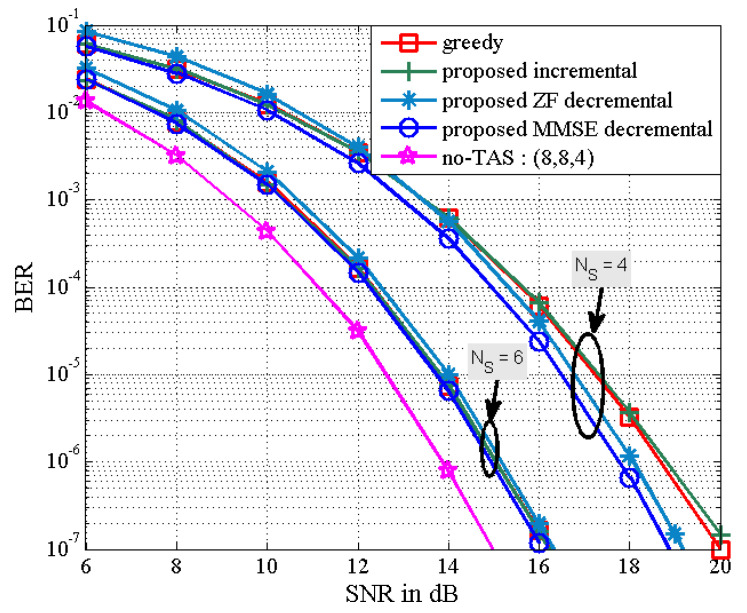
BER the proposed TAS selection algorithms for $\left(8,N_{S},4\right)$ MU–STLC system.

**Figure 6 sensors-21-02690-f006:**
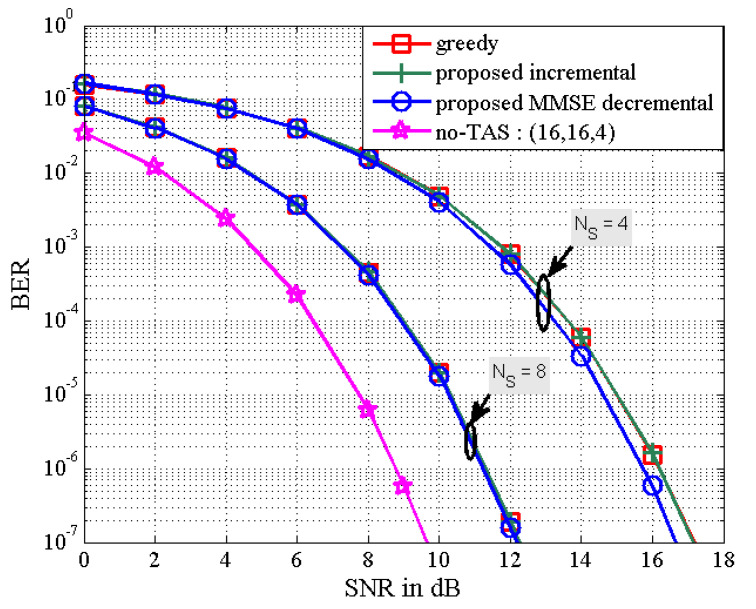
BER the proposed TAS selection algorithms for $\left(16,N_{S},4\right)$ MU–STLC system.

**Figure 7 sensors-21-02690-f007:**
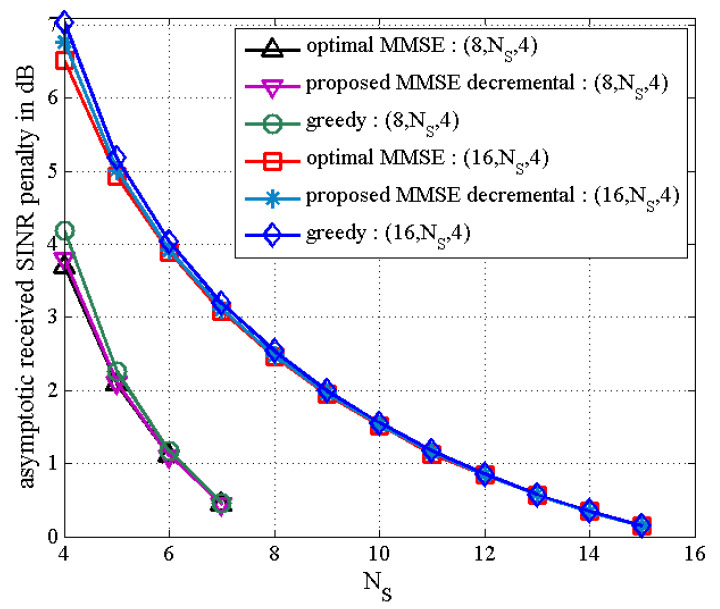
Received signal-to-interference-plus-noise ratio (SINR) loss for MU–STLC system with TAS selection for $\left(8,N_{S},4\right)$ and $\left(16,N_{S},4\right)$.

**Figure 8 sensors-21-02690-f008:**
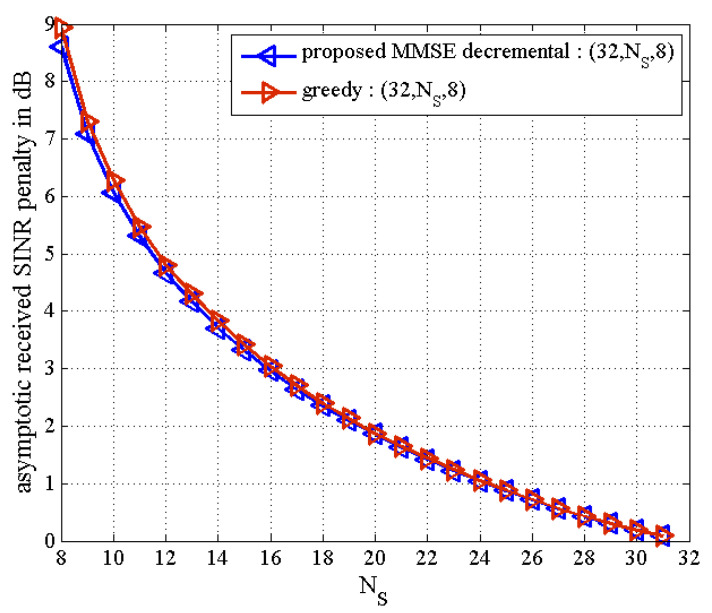
Received SINR loss for MU–STLC system with TAS selection for $\left(32,N_{S},8\right)$

**Figure 9 sensors-21-02690-f009:**
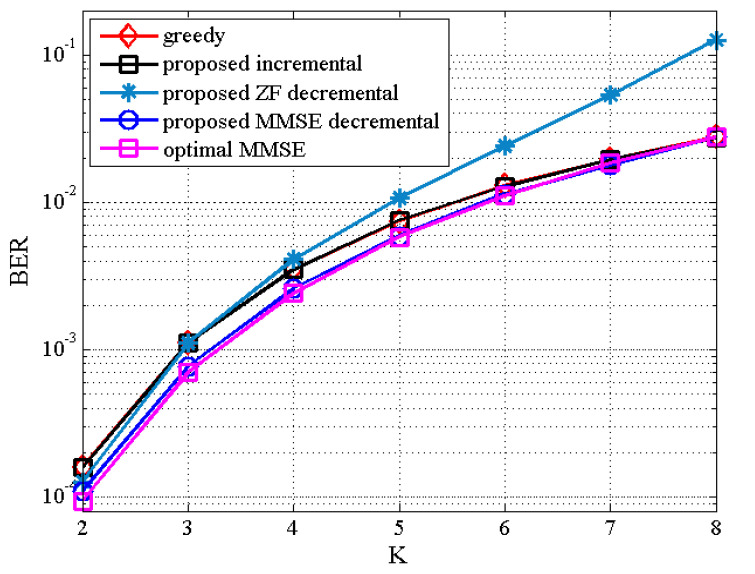
BER the proposed TAS selection algorithm for $\left(8,N_{S},K\right)$ MU–STLC system with $N_{S}=K$ under SNR = 12 dB.

**Figure 10 sensors-21-02690-f010:**
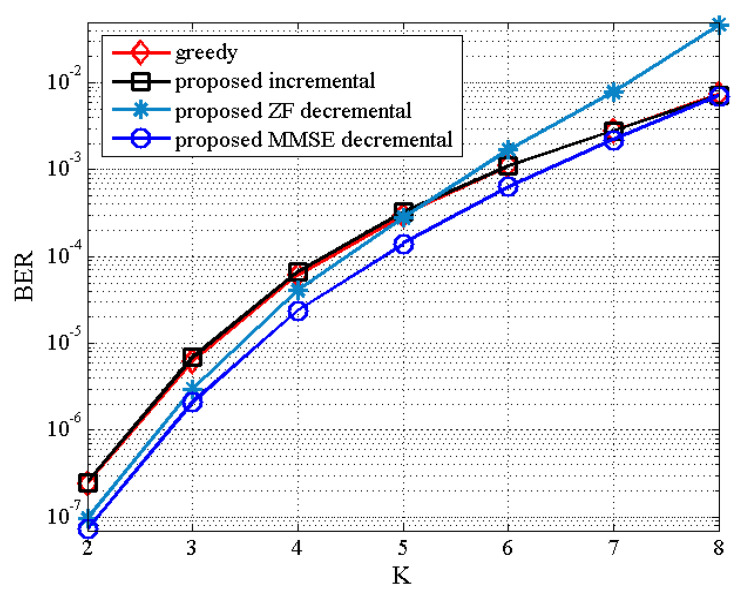
BER the proposed TAS selection algorithm for $\left(8,N_{S},K\right)$ MU–STLC system with $N_{S}=K$ under SNR = 16 dB.

**Table 1 sensors-21-02690-t001:** Computational complexity of complex multiplications and complex summations in three matrix-related operations.

Expressions	CMs	CSs
AB	NP​M	NP(M−1)
$D=AA^{H}$	0.5MN(N+1)	0.5(M−1)N(N+1)
$D^{-1}$	$0.5N^{3}+1.5N^{2}$	$0.5N^{3}-0.5N^{2}$

## Data Availability

Not applicable.

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
