# Peer review of "Efficient Transmit Antenna Subset Selection for Multiuser Space–Time Line Code Systems"

_sensors, 2021, doi:10.3390/s21082690_

Round 1
Reviewer 1 Report
The author tackles the problem of the efficient transmit antenna subset selection for maximizing the signal-to-interference-plus-noise ratio of multiuser space-time line code systems. This is a interesting piece of work, however, I've got some comments on it.
1) You should clearly state the novelty of your paper. In its actual form, it is not that clear what you have proposed.
2) There is no comparisons with related work. How does your proposed scheme compare with the state of the art? You should add a section contrasting your work with works related to yours.
3) The number of references should be increased. That can be done by adding a related work section.
4) You should improve the description of the algorithms listed on your manuscript. It feels like you could improve on the discussions about the algorithms you provided.
5) Improve the quality of all figures. They all seem blured.
6) Add grid line to figures 1 and 2.
7) Add more points to greed, proposed and optimal in Figure 7. For those lines,Ns stops at 7. You should try to improve the results.
8) The discussions on results is too poor. You should enhace the discussion and analysis of the important results you provide on your manuscript.
Reviewer 2 Report
The paper is interesting, dealing with an actual topic on optimal antenna selection in multiuser space-time line code.
Two algorithms are proposed in order to reduce the search effort replacing an exhaustive search.
The simulations results show the performances of proposed algorithm in terms of bit error rate.
For the proposed algorithms the complexity analysis is performed, and complexity comparison prove the algorithm efficiency especially for large number of selected antennas.
Reviewer 3 Report
The paper proposes algorithms for incremental and decremental selection of transmit antenna subset (TAS) with low complexity for multiuser space-time line code (MU-STLC) systems. It is shown that the proposed algorithms are less complex than the existing algorithms for MU-STLC systems. This is especially evident for a large number of transmitting antennas. The complexity of the algorithms is estimated. It is shown that the complexity of the algorithms is much lower than that of other algorithms, especially for large NS. The results of analyzes of performance versus size of antenna systems are presented.
The work consists of an introduction, conclusion and several sections. The MU-STLC model with TAS selection is described in Section 2. Section 3 proposes algorithms for incremental and decremental TAS selection. Section 4 analyzes the computational complexity of the proposed algorithms. Section 5 compares the performance of the proposed algorithms and other methods.
Suggestions for improving the manuscript:
- It is desirable to expand the review and add works on the selection of antennas from the journals of the MDPI publishing house.
- Replace the designation for the matrix in the formula (13) on the left side, for example, with V or another letter. Such designations make it difficult to read.
- The formulas for estimating the complexity of algorithms (51), (52), (53), (54), (55), and (56) need to be refined. For example, the expression for (53) can be refined as N= (2 N_t – N_s + 1) / 2 * N_s * (48 K^2 + 76 K + 40).
